# *Drosophila* Model for Studying Gut Microbiota in Behaviors and Neurodegenerative Diseases

**DOI:** 10.3390/biomedicines10030596

**Published:** 2022-03-03

**Authors:** Meng-Hsuan Chiang, Shuk-Man Ho, Hui-Yu Wu, Yu-Chun Lin, Wan-Hua Tsai, Tony Wu, Chih-Ho Lai, Chia-Lin Wu

**Affiliations:** 1Graduate Institute of Biomedical Sciences, Department of Biochemistry, Department of Microbiology, Department of Biomedical Sciences, College of Medicine, Chang Gung University, Taoyuan 33302, Taiwan; tony12185@gmail.com (M.-H.C.); unaho1122@gmail.com (S.-M.H.); winney0614@gmail.com (H.-Y.W.); lyc2001bs@gmail.com (Y.-C.L.); 2Research and Development Department, GenMont Biotech Incorporation, Tainan 74144, Taiwan; twh@genmont.com.tw; 3Department of Neurology, Molecular Infectious Disease Research Center, Department of Pediatrics, Linkou Chang Gung Memorial Hospital, Taoyuan 33305, Taiwan; tonywu@cgmh.org.tw; 4Department of Neurology, New Taipei Municipal Tucheng Hospital, Tucheng 23652, Taiwan; 5Department of Neurology, Xiamen Chang Gung Hospital, Xiamen 361028, China; 6Department of Medical Research, Graduate Institute of Biomedical Sciences, China Medical University and Hospital, Taichung 40402, Taiwan; 7Department of Nursing, Asia University, Taichung 41354, Taiwan; 8Brain Research Center, National Tsing Hua University, Hsinchu 30013, Taiwan

**Keywords:** *Drosophila melanogaster*, microbiota, gut−brain axis, behaviors, learning and memory, neurodegenerative diseases

## Abstract

Mounting evidence indicates that the gut microbiota is linked to several physiological processes and disease development in mammals; however, the underlying mechanisms remained unexplored mostly due to the complexity of the mammalian gut microbiome. The fruit fly, *Drosophila melanogaster*, is a valuable animal model for studying host-gut microbiota interactions in translational aspects. The availability of powerful genetic tools and resources in *Drosophila* allowed the scientists to unravel the mechanisms by which the gut microbes affect fitness, health, and behavior of their hosts. *Drosophila* models have been extensively used not only to study animal behaviors (i.e., courtship, aggression, sleep, and learning & memory), but also some human related neurodegenerative diseases (i.e., Alzheimer’s disease and Parkinson’s disease) in the past. This review comprehensively summarizes the current understanding of the gut microbiota of *Drosophila* and its impact on fly behavior, physiology, and neurodegenerative diseases.

## 1. Introduction

The fruit fly, *Drosophila melanogaster*, is an important experimental animal used in biomedical studies. The fruit fly was first used as an experimental organism by Castle in 1901 and Morgan in 1909 for genetic studies. Since *Drosophila* has been used as a model organism for over a century, several techniques have been developed to address specific scientific questions. *Drosophila* has a relatively short lifespan and complex behaviors. The availability of powerful tools for genetic manipulation in specific tissues or organs allowed the scientists to study the molecular mechanisms regulating *Drosophila* behavior, such as adaptive response to the surrounding environment, as well as innate (courtship, aggression, sleep, etc.) [1,2,3] and learned behaviors (olfactory memory and courtship memory) [4,5,6].

Over the past 20 years, *Drosophila* has been used as an ideal in vivo model for studying neurodegenerative diseases, such as Alzheimer’s disease (AD) and Parkinson’s disease (PD) [7,8]. The prominent signaling pathways that regulate neuronal growth and functions in humans are conserved in *Drosophila*, allowing to use of this model organism for studying human neurodegenerative diseases. For example, the *Drosophila* model of AD developed by ectopically expressing human Aβ42 protein in the fly brain exhibits clinical symptoms associated with AD patients, such as age-dependent short-term memory impairment, learning defects, increased wakefulness, and sleep disruption [8,9]. This suggests that *Drosophila* can be used as an ideal in vivo model for screening novel drugs for the treatment of AD.

Mounting evidence supports the notion that a well-balanced gut microbiota is essential to human health [10,11]. Interfering with the composition of the gut microbiota may adversely affect the host health and physiology, leading to various neurological diseases, such as behavioral abnormalities, neurodegenerative diseases, and neuropsychiatric disorders [12,13,14]. Studies showed that the gut bacteria can produce several neurotransmitters, including dopamine, gamma-aminobutyric acid (GABA), and serotonin [15]. Moreover, the bidirectional crosstalk between the intestine and the central nervous system (CNS) is mediated through the gut microbiota [16,17]. Recent studies have indicated that alterations in the gut microbiota composition are related to certain neurological diseases [18,19,20]. However, the detailed mechanism of how intestinal bacteria regulate disease progression by influencing the communication along the gut-brain axis requires further investigation.

Over the past 20 years, most of the gut microbiota studies employed rodent models to investigate the role of the microbiota−gut-brain axis. However, the versatile organism, *Drosophila*, is a powerful research model that allows scientists to investigate the mechanisms by which gut microbes affect the brain functions (Figure 1a,b). The microbes can be genetically tagged with a fluorescent protein, thus, their distribution in the digestive tract of *Drosophila* can be visualized (Figure 1c) [21,22].

## 2. *Drosophila* Innate Behaviors

Innate behaviors are believed to be hardwired in the animal brain and genetically encoded, and require no prior experience for proper execution. Innate behaviors are highly adaptable to the internal state of animals and the external environment, which allows animals to modify their behaviors according to different environmental conditions [23]. *Drosophila* is an excellent animal model for studying the molecular mechanisms of human innate behaviors, since there are some similarities of behavioral phenotypes between flies and human. In *Drosophila*, a variety of innate behaviors have been studied, such as mating behavior, aggressive behavior, sleep/locomotor activity, and temperature-related behavior [3,24,25,26,27,28]. In addition, the *Drosophila* genome is around 60% homologous to that of human and approximately 75% of the human disease-related genes also have homologs in flies [29]. For example, in sleep behavior, the dopamine signaling regulates the wakefulness not only in *Drosophila*, but also in humans [30]. Herein, we introduce some crucial *Drosophila* innate behaviors, and discuss their relationship with the gut microbiota (Table 1).

### 2.1. Gut Microbiota and Mating Behavior

Mating behavior in male *Drosophila* includes a series of steps: progression towards the female (orientation), chasing the female (following), touching the female’s abdomen (tapping), unilateral wing extension and vibration (singing), licking the female’s genitalia, grasping up to the female’s back (attempted copulation), and final copulation [3,26]. One male fly attempts to mate with one virgin female, and the measurement of the male fly’s sexual enthusiasm towards the female is called the courtship index (CI). CI is the time spent by the male *Drosophila* on courtship divided by the total time of the experiment. The well-known neural circuit that regulates mating behavior is composed of *fruitless* (*fru*)-expressing neurons. The *fru*-expressing neurons including primary sensory afferents and central interneurons, such as P1 cluster neurons that trigger courtship, may form the core portion of the male courtship circuit [50].

At least three studies have reported that the gut microbiota does not affect courtship behavior in *Drosophila* [33,35,51]; however, it should be noted that each study used different wild-type flies and employed different methods to generate germ-free flies. Contrary to these studies, Heys et al. showed that the gut microbiota counteracts the male outbreeding strategy by altering the female sexual signaling, which leads to reduced sperm transfer from male flies [31]. Outbreeding in *Drosophila* increases genetic diversity in the offspring, enhances the immune defense, and may be affected by the host microbiota [31,52]. A previous study showed that 1- and 3-days post-mating, the gene expression pattern in the fly head tissue is altered, which affects fly behavior, including feeding behavior, metabolism, and egg production (post-mating behavior) [53]. Delbare et al. performed a large-scale RNA-seq analysis of virgin and mated axenic/conventional flies, and reported that, in mated females, the transcripts involved in regulating reproduction and neuronal functions are differentially abundant depending on the microbiota of females [32]. Moreover, the microbiota status of the male affected fecundity in both conventional and axenic females; however, it affected the transcriptional profiles only in axenic, but not in conventional, females.

### 2.2. Gut Microbiota and Aggressive Behavior

Aggressive behavior, which is widely observed in the animal kingdom, is critical for survival and reproduction [54]. In *Drosophila*, aggressive behavior includes wing threat, boxing, tapping with the front leg, lunging, and head butting [2,33]. Jia et al. described a protocol to analyze aggressive behavior in flies. In this assay, two male or female flies (5–7 days old) are placed in a fighting chamber, which is then rapidly covered with a 20 mm × 20 mm glass coverslip. The fight during the 30 min is recorded using a camera. Aggressive behavior is analyzed by counting the number of lunges in males and head butting in females, and evaluating the lunging frequency/fighting latency. Lunging frequency is defined as the number of lunges per minute after lunging initiation, whereas fighting latency is defined as the duration between the beginning of the recording and the first lunging [33]. Studies suggest that aggressive behaviors coordinate with visual, olfactory, gustatory, mechanosensory, and auditory inputs. These inputs integrate into the brain mushroom body (MB), lateral horn, and central complex regions, and finally drive the motor outputs through the wing and leg muscles. Studies also suggest that neurotransmitters and neuropeptides, such as octopamine, serotonin, dopamine, acetylcholine, and neuropeptide F (NPF), also play important roles in aggressive behaviors [54].

Recent studies have shown that specific gut microbiota is associated with aggressiveness in flies. Jia et al. used sodium hypochlorite and a series of wash protocols to generate axenic flies [33]. Compared to conventional flies (Canton-S), axenic flies showed less inter-male aggressive behavior, indicating that germ-free flies have reduced inter-male aggression. Furthermore, commensal bacteria such as *Acetobacter*, *Lactobacilli*, and *Enterococci* promoted aggressive behaviors in axenic flies by microbial re-colonization. The authors also reported a 73% reduction in octopamine levels in germ-free flies, and genetic manipulation of the octopaminergic neuronal activity restored aggressive behavior in axenic flies. The same study also reported two key factors involved in regulating aggression via octopamine expression. One was the critical developmental period during the larval stage (48–98 h after egg laying), in which supplementation with yeast-rich diet restored aggressive behavior in axenic flies. The other factor was nutrition, as it is also important for octopamine expression, and providing proper nutrition to axenic flies normalized the aggressive behavior. These findings demonstrate that gut microbiota and nutrition during the critical developmental period affect octopamine expression, which regulates aggression in adult males [33].

### 2.3. Gut Microbiota and Sleep and Locomotor Activity

The *Drosophila* Activity Monitor (DAM) system is used to study sleep and locomotion behavior in flies [55]. In this assay, flies are housed under 12 h light/12 h dark cycle conditions [56,57]. The DAM apparatus has three major components: (1) a glass tube in which a single fly can move freely, (2) an infrared beam which is directed through the midpoint of the glass tube, and (3) a computer for recording the behavioral results [55]. In the sleep/locomotor behavior assay, the DAM system records infrared beam breaks due to flies walking in glass tubes to detect sleep or locomotion [58]. The well-known sleep behavior-related circuit in the *Drosophila* brain is composed of circadian neurons. Light stimulates a group of large ventral−lateral neurons (l-vLNs), which release pigment dispersing factors (PDFs) to stimulate a group of small ventral−lateral neurons (s-vLNs). Finally, the s-vLNs regulate the brain locomotor control center, the ellipsoid bodies, to inhibit sleep behavior. Both l-vLNs and s-vLNs express GABA_A_ receptors, which might inhibit wakefulness through sleep-promoting GABAergic neurons. MB is also an important brain region that regulates sleep via the catalytic subunit of protein kinase A, an enzyme activated by the sleep-relevant cAMP pathway [59]. Locomotor activity is important for *Drosophila* survival, and is controlled by hormones, which can also be analyzed using the DAM system. King et al. recently reported the presence of a peptidergic circuit that links the circadian clock to locomotor activity. s-vLNs release PDFs that stimulate DN1 clock neurons which, in turn, stimulate *DH44* positive pars intercerebralis (PI) neurons. *DH44* positive PI neurons stimulate hugin-positive SEZ neurons, which finally release the hugin neuropeptide to the motor circuit in the ventral nerve cord that regulates locomotor activity in *Drosophila* [60].

The effect of the gut microbiota on sleep behavior in *Drosophila* remains unclear. Selkrig et al. used the isogenic *Wolbachia*-free *D. melanogaster* Canton-S strain as the wild-type flies, and performed sleep and locomotor behavior assays. The authors generated *Wolbachia*-free Canton-S germ-free flies by washing the eggs with NaClO, and termed the first generation as F1 and the next generation as F2. Their data revealed that neither germ-free F1 nor germ-free F2 flies exhibit any significant change in sleep behavior. Moreover, even though the germ-free F1 flies did not showed any change in the locomotor behavior, germ-free F2 flies exhibited a mild increase in the locomotor activity compared to conventional flies [35]. Another study by Silva et al. employed Canton-S and W^1118^ as wild-type flies, and used NaClO to wash the fly eggs for 3–5 min to generate axenic flies [61]. Their results revealed that compared to conventional flies, axenic flies tended to sleep for a longer duration and displayed reduced sleep rebound after sleep deprivation [34]. However, the gut microbiota status of the flies had a small effect on the circadian rhythmicity of the locomotor activity.

Silva et al. and Selkrig et al. focused on understanding the role of the gut microbiota in regulating *Drosophila* sleep and locomotion. Schretter et al. proposed a possible mechanism to explain the relationship between the gut microbiota and locomotion/sleep. In this study, Oregon-R strain was used as the wild-type, and the flies received the antibiotic-supplemented diet to generate antibiotic-treated (ABX) flies. Results showed that ABX flies have increased walking speed, enhanced daily activities, and reduced sleep during the daytime. The authors also reported that the enzyme xylose isomerase, which catalyzes the reversible isomerization of certain sugars present in *Lactobacillus brevis*, reduces locomotor activity by modulating sugar metabolism via stimulation of the octopaminergic neurons; however, this enzyme does not affect sleep [36].

### 2.4. Gut Microbiota and Temperature Related Behaviors

Environmental temperature strongly affects the physiology of ectotherms, such as *Drosophila*. Temperature preference behavior (flies avoid hot or cold environments) and tolerance towards hot or cold temperatures are important for survival in *Drosophila*. Temperature gradient assay is used to study temperature preference in flies. Different studies used distinct temperature gradient devices, but the basic principle is the same: one side of the device is at low temperature, and the other side produces high temperature. By placing an aluminum sheet between the two sides, a temperature gradient is produced. Flies are placed on the aluminum sheet and are allowed to choose their preferred temperature [62,63]. There are two different ways to study temperature avoidance behavior: one is to use the temperature gradient assay (described above), and the other is using the two-choice assay machine. The two-choice assay machine contains four quadrants that are built with a thermoelectric cooling chip and an aluminum sheet. When performing the temperature avoidance assay, the two-choice assay machine would set a constant temperature (25 °C) in one set of diagonal quadrants, and the other diagonal quadrants were set at the experimental temperature, which depends on the analysis [64]. Chill coma recovery is protocol to study temperature tolerance behavior. In this assay, flies are immersed into a water−glycol bath set at −3.5 °C (depending on the experimental design) for cold stress or exposed to 38.5 °C temperature (depending on the experimental design) for heat stress, following which the flies are allowed to recover and the recovery time is measured [37,65].

Recent studies have suggested that the gut microbiota strongly affects temperature tolerance in fruit flies. Henry et al. generated axenic flies by dechorionation, in which the fly eggs were immersed in 2.7% hypochlorite for 2 min, 70% ethanol for 2 min, and finally washed twice in autoclaved water [37]. After dechorionation, the flies were reared on autoclaved food. Their results suggest that the cold tolerance of axenic flies is defective compared to that of conventional flies. However, heat tolerance was not affected in the axenic flies. This suggests that microbiota disruption decreases the ability to cope with cold stress [37]. Studies on other *Drosophila* species, such as *D. subobscura*, suggest that conventional flies exposed to mild heat conditions (<34 °C) exhibit higher thermal tolerance than axenic flies [38].

## 3. *Drosophila* Learned Behaviors

Tremendous research progress has been made over the past 40 years on detailed mechanisms underlying *Drosophila* learning and memory. Several *Drosophila* learning and memory behaviors have been used in biological studies, including olfactory memory, visual memory, place memory, and courtship memory [66]. Ample studies have suggested that flies can learn and form memories from their experiences, and these memories can persist for hours or days. *Drosophila* olfactory memory and courtship memory are two popular assay systems for memory studies in the laboratory. Herein, we mainly focus on olfactory and courtship memories in *Drosophila*.

### 3.1. Olfactory Memory-Aversive and Reward Conditionings

*Drosophila* olfactory memory was described by Tempel et al. [4] and Tully et al. [5]. For aversive olfactory conditioning, flies receive the electric shock punishment along with exposure to a specific odor (CS+), followed by a different odor (CS−) without electric shock. For memory testing, flies are transported using a small elevator in the training machine and allowed to select between two odors (CS+ vs. CS−) in the T-maze. Both 3-octanol (OCT) and 4-methylcyclohexanol (MCH) are often used in this assay. The performance index is defined as:$$\frac{\left(\mathrm{number}\;\mathrm{of}\;\mathrm{flies}\;\mathrm{in}\;\mathrm{the}\;\mathrm{CS}+\;\mathrm{odor}\;\mathrm{arm}-\mathrm{number}\;\mathrm{of}\;\mathrm{flies}\;\mathrm{in}\;\mathrm{the}\;\mathrm{CS}\;-\mathrm{odor}\;\mathrm{arm}\right)}{\mathrm{total}\;\mathrm{number}\;\mathrm{of}\;\mathrm{flies}}\times 100\;$$

A performance index of a higher value indicates that flies have greater memory abilities.

The sugar-reward memory assay, described in earlier studies, is similar to the aversive olfactory memory assay, and the water-reward memory assay is a modification from the sugar-reward memory [4,67,68,69,70]. In sugar-reward and water-reward memory assays, hunger or thirsty flies are trained to pair an odor (CS+) with sugar or water, respectively. Before starting the experiment, flies are transferred to vials containing a moist filter paper for food starvation before the sugar-reward memory assay, or to vials containing dry sucrose filter paper for desiccation before the water-reward memory assay. For training, flies are moved to a training tube containing a dried filter paper and exposed to the first odor (CS−). The flies are then transferred to another training tube containing either a sucrose filter paper or a water filter paper and then exposed to the second odor (CS+). For testing, flies are allowed to choose between the two odors in a T-maze, and their distribution in the tubes was analyzed and presented as the performance index [68,71].

### 3.2. Courtship Memory

Already mated female flies reject male copulation attempts, which eventually leads to reduced courtship behavior of male flies. This unpleasant experience (rejection from female) also suppresses the male courtship behavior towards a virgin female [6]. The suppression of the male fly courtship behavior towards a virgin female following exposure to already mated females is defined as courtship conditioning. Before testing, naïve males are collected on the day of eclosion, and kept individually for 4–5 days. When naive males are sexually mature, they are exposed to a mated female for 1 h, and the courtship index during the first (CI_initial_) and the last (CI_final_) 10 min of the 1 h experimental period was calculated. The test male is then moved to a new place and exposed to a decapitated virgin female for 10 min (CI_test_). In the control group, naive male flies are kept alone in a chamber for 1 h, and then transferred to a new chamber with a decapitated virgin female for 10 min (CI_sham_) [72]. Finally, the courtship learning index and memory index are calculated. Lower learning and memory indices indicate that male flies have better learning and memory ability. Learning index = CI_final_/CI_initial_ and memory index = CI_test_/mean CI_sham_ [72].

### 3.3. Impact of the Gut Microbiota on Memory

Studies have shown that flies can be trained to associate an odor with punishment or reward [4,5,67,73]. For olfactory memory, projection neurons in the antennal lobe receive inputs from olfactory receptor neurons, and project to the lateral horn and MBs in the fly brain [74]. Notably, *Drosophila* MBs are critical for olfactory memory [75,76]. MB neurons are classified as αβ, α′β′, and γ neurons [77]. Blum et al. showed that in flies, learned experiences form short-term memory in the MB γ neurons, while long-term memory is formed in the MB αβ neurons [78]. In contrast, McBride et al. reported that MBs are essential for the consolidation of long-term memory during courtship conditioning [79]. Another study reported that the overlap of MB neurons is required for courtship memory and olfactory associative memory in *Drosophila* [80]. Growing evidence suggests that the gut microbiome may have some effects on animal behavior [81], and the interaction between the gut and the brain is critical for behaviors [82]. Several researchers have indicated a link between the gut microbiota and learning [83] or memory [84]. In particular, specific probiotics have been developed to improve learning and memory in mice [85,86]. In *Drosophila*, the gut microbiota is much simpler compared to other animals [87]. Fruit flies are a potential animal model to study the effects of the gut microbiota on learned behaviors; however, there is only scattered evidence. Recently, Silva et al. reported that axenic flies have reduced appetitive memory at 24 h after training, and axenic male flies exhibit decreased courtship learning and memory [34](Table 1). Further studies are needed to elucidate the impact of the gut microbiota on learned behaviors in the future.

## 4. Aging and Longevity

Aging is a universal biological phenomenon. Essentially, aging is a term used to describe the relationship between advancing chronological age and decline in physiological functions, such as reproductive function, muscle strength, cognitive impairment, and age-related diseases [88]. Several studies have employed animal models, such as *Caenorhabditis elegans*, *Drosophila*, and mice, to characterize the molecular and physiological changes that occur during aging [89].

### Gut Microbiota and Drosophila Aging

Aging is associated with an increased risk of chronic diseases, including immune system diseases, cancer, neurodegeneration, and dysbiosis [90]. A recent study demonstrated that there are significant differences in the gut microbiota composition between healthy and unhealthy elderly adults (aged >90 years). Healthy elderly people have a higher abundance of *Bacteroidetes* in the gut, which provide more functional pathways for energy metabolism. However, unhealthy elderly individuals have a higher abundance of *Streptococcus* and other pathogenic bacteria in the gut [91]. Studies on the elderly human microbiota are not very consistent due to different internal and external parameters associated with each person; therefore, further studies using animal models are required to clarify the role of the gut microbiota in aging.

Typically, the healthy and well-maintained wild-type *Drosophila*, W^1118^, has a median lifespan of approximately 70 days, and a maximum of approximately 90 days when reared at 25 °C [92]. Aged flies exhibit serious age-related behavioral changes 21 days after eclosion, such as age-related locomotor impairment (ARLI), decreased sexual activity and reproductive behavior, reduced cold temperature sensing ability, breakdown of sleep−wake cycles, and impaired cognitive functions [63,93,94,95]. These age-related behavioral changes are associated with a change in the gene expression pattern. Aging is associated with changes in the expression of nearly 23% of genes in *Drosophila*, and many of these genes are related to stress response and reproduction. Perturbed expression of these genes causes impairment of several physiological processes in aged flies [96].

In adult flies, *Proteobacteria* (*Acetobacter* and *Komagataeibacter*) and *Firmicutes* (*Lactobacillus* and *Leuconostoc*) comprise a major part of the gut microbiota, especially in the stomach-like copper cell region (CCR, also see the Figure 1a), which controls the distribution and composition of the microbiota and metaplasia of the gastric epithelium via activation of the JAK/STAT signaling in the aging gut [97]. Moreover, there is a significant difference in the gut microbiota composition between young and old flies. In young flies, *Acetobacter persici* and *L. brevis* are the dominant species. However, *A. malorum* and *L. plantarum* are mainly present in the gut of old flies [98]. Studies elucidating the effects of the gut microbiota during different stages of *Drosophila* development are not consistent. Brummel et al. found that flies reared under axenic conditions have reduced lifespan compared to conventional flies. Furthermore, the presence of microbiota during the first week of adult life in *Drosophila* increases the lifespan. In contrast, the presence of microbiota during the later life decreases lifespan. The authors proposed that the gut microbiota interacts with longevity genes to modulate lifespan in *Drosophila* [39]. Contrary to this, Catterson et al. showed that intermittent fasting (IF, two days of feeding followed by five days of fasting) can prolong the lifespan in *Drosophila.* Further, the authors showed that IF for at least 30 days during adulthood enhances lifespan. The authors argued that IF increases starvation-induced resistance to oxidative and xenobiotic stress. Furthermore, IF enhances the lipid content in 60-day-old flies. Analysis of the gut 40 days post-IF revealed a significant reduction in age-related pathologies, improved gut barrier function, and reduced bacterial abundance including that of *L. plantarum*. IF can act independently of the TOR pathway and robustly increase the lifespan. Altogether, during the early life, short-term IF can induce long-lasting beneficial effects which could enhance longevity, at least in part, by preserving the gut health [40].

## 5. Alzheimer’s Diseases

Alzheimer’s disease (AD) arises due to neuronal degeneration in the brain. AD is characterized by the presence of neuritic plaques and neurofibrillary tangles, which leads to amyloid-beta (Aβ) peptide accumulation [99]. There are two major hypotheses to explain the mechanism underlying AD development. The first hypothesis is the cholinergic hypothesis. In the 1970s, studies revealed that neocortical and presynaptic cholinergic deficits are related to choline acetyltransferase (ChAT), an enzyme involved in acetylcholine (ACh) synthesis. Since ACh is the key factor involved in regulating cognitive functions, the cholinergic hypothesis proposes that β-amyloid accumulation affects cholinergic neurotransmission, and causes a reduction in choline uptake and Ach release [100,101]. The second hypothesis is the amyloid hypothesis. It has been observed that Aβ degradation is reduced with increased age or in various pathological conditions, leading to the accumulation of Aβ peptides, which finally causes neuronal death [102,103,104]. Several AD risk factors have been identified, such as aging, genetic background, amyloid precursor protein mutations, environmental factors, and heavy metal exposure [105]. 

### 5.1. Gut Microbiota and AD in Drosophila

AD patients exhibit reduced abundance of *L. brevis* and *Bifidobacterium dentium* in the gut and decreased GABA levels in the CNS [106]. Serotonin, another key neurotransmitter that regulates cognitive functions, can be synthesized by the gut microbes [15,107]. Besides neurotransmitter imbalance, neurotoxins are also implicated in AD. A recent study revealed that the neurotoxin β-N-methylamino-L-alanine (BMAA) is produced by *Cyanobacteria* in the gut, which contributes to AD development [106,108]. However, the impact of gut microbiota on neural circuits and related mechanisms in AD remain unclear. Overall, three *Drosophila* AD models have been developed that allowed scientists to uncover the relationship between AD and gut microbiota. These include *elav-Gal4*, *UAS-BACE/UAS-APP* model, *GMR-A**β42* model, and *elav-Gal4*; *UAS-A**β42* model (Table 1).

### 5.2. elav-Gal4; UAS-BACE/UAS-APP Fly Model

The *UAS/Gal4* system has been used to develop AD fly models by expressing human beta-secretase (BACE) or amyloid precursor protein (APP) under the control of *elav-Gal4* driver, which is expressed specifically in neurons. These flies exhibit increased amyloid deposition in neurons, loss of climbing ability, and increased neurodegeneration. A recent study further showed the association between Kefir uptake and AD in *elav-Gal4*; *UAS-BACE/UAS-APP* flies. Kefir is a probiotic that contains five major bacterial species: *L. kefiranofaciens* (21.96%), *L. kefiri* (0.2%), *A. fabarum* (0.17%), *Lactococcus lactis* (0.004%), and *Rickettsiales* (0.001%). The major metabolites present in Kefir are hexane (Hex), dichloromethane (DCM), ethyl acetate (EtOAc), and n-butanol (But-OH). Kefir administration to AD model flies enhanced their climbing ability, improved the survival rate, and suppressed neurodegeneration. Furthermore, although the four metabolites present if Kefir were individually able to improve the climbing activity in AD flies, only EtOAc and But-OH were able to extend the lifespan of the AD model *Drosophila* [41].

Westfall et al. developed a symbiotic formulation containing *L. plantarum* NCIMB 8826 (Lp8826), *L. fermentum* NCIMB 5221 (Lf5221), and *Bifidobacteria longum* spp. *infantis* NCIMB 702255 (Bi702255), along with 0.5% TFLA (polyphenol plant extract from the gastrointestinal tonic Triphala) powder. Treating *elav-Gal4; UAS-BACE/UAS-APP* flies with this symbiotic formulation reduced the expression of *Drosophila* insulin-like peptide (Dilp) including Dilp2 and Dilp3, and Dilp receptors, whereas it increased dFOXO and innate immune factor *dual oxidase* expression, and promoted IMD signaling. Moreover, the symbiotic formulation reduced the levels of total oxidants, suppressed lipid peroxidation (LPO), and restored the activity of the mitochondrial electron transport chain (ETC) complexes in AD model flies [42].

### 5.3. GMR-Aβ42 Fly Model

GMR-Aβ42 AD fly model was developed by expressing the human Aβ42 transgene under the control of the glass multimer reporter (GMR)-*Gal4* driver, which is expressed in retinal cells. These flies exhibit a rough eye phenotype due to neurodegeneration, and this model is widely used to investigate mechanisms underlying AD development [109]. Liu et al. showed that *L. sakei* Probio 65 and *L. paracasei* 0291 feeding significantly reduced AD-associated neurodegeneration in GMR-Aβ42 AD model flies [43]. Moreover, Tan et al. reported that *Lactobacillus* rescues the rough eye phenotype in AD flies, especially the *L. plantarum* DR7 strain. *L. plantarum* DR7 can restore the gut microbiota diversity in AD model flies by increasing the abundance of *Stenotrophomonas* and *Acetobacter*, while reducing that of *Wolbachia* [44].

### 5.4. elav-Gal4; UAS-Aβ42 Fly Model

These AD model flies were developed by expressing human *Aβ42* under the control of the pan-neuronal driver *elav-Gal4*. Similar to other AD model flies, *elav-Gal4*; *UAS-Aβ42* flies exhibit decreased survival rates and reduced climbing abilities. Moreover, the abundance of families *Acetobacteraceae* and *Lactobacillaceae* is decreased in the gut of these flies, especially *Acetobacter* and *Lactobacillus.* Using GC-MS, it has been found that levels of acetate, which is the most abundant short-chain fatty acid (SCFA) in flies, are decreases dramatically in AD model flies [45]. In vitro and in vivo studies have demonstrated that SCFAs play a potential role in reducing Aβ levels and mitigating its toxic effects [110,111]. Furthermore, enteric dysbiosis caused due to infection with a nonpathogenic *enterobacteria* (Ecc15) in adult flies promotes hemocyte recruitment to the brain, leading to TNF−JNK-mediated neurodegeneration, resulting in AD development [46]. 

## 6. Parkinson’s Disease

Parkinson’s disease (PD) is a neurological disorder that affects people mainly in later years of life. In general, PD is rarely observed in those under 40 years of age, and it is estimated that about 1% of people over 60 years of age and 0.3% of the entire population develop this disease [112,113,114]. The accumulation of α-synuclein-containing Lewy bodies and loss of dopaminergic neurons in the substantia nigra are the major pathological changes observed in PD patients. The motor features commonly observed in PD patients include tremors, rigidity, and bradykinesia. Furthermore, common non-motor features observed in patients with PD are sleep dysfunction, autonomic dysfunction, and hyposmia [115]. Although the causes underlying PD development are still unknown, the major risk factors include age, environmental factors, and genetics.

### 6.1. Gut Microbiota and PD in Drosophila

It has been reported that in patients with PD, the abundance of *Enterobacteriaceae* in the gut is positively associated with the severity of postural instability [46]. Furthermore, it has been reported that, in PD patients, the gut abundance of *Prevotellaceae*, *Blautia* spp., *Coprococcus* spp., *Roseburia*, *Faecalibacterium* spp., and *Prevotella* spp. Is decreased, while that of *E. coli*, *Ralstonia*, *Lactobacillus*, *Bifidobacteriu*, *Verrucomicrobiaceae*, *Bacteroides*, *Parabacteroides*, *Akkermansia*, *Butyricimonas*, *Veillonella*, *Odoribacter*, *Mucispirillum*, and *Bilophila* is increased [12,116,117,118,119]. However, the effects of the gut microbiota on PD pathogenesis remain unclear. The *Drosophila* PD model has been used to investigate the relationship between PD and gut microbiota. Similar to the AD model, there are two model systems to study PD in *Drosophila*: *elav-Gal4*; *UAS-Synuclein* model, and the PINK1 mutant model (Table 1), as described below.

### 6.2. elav-Gal4; UAS-Synuclein Fly Model

In PD patients, there is a multiplication of or mutations (A53T, A30P, or E46K) in the α-synuclein gene. Fly PD models have been developed by expressing the mutant α-synuclein under the control of *elav-Gal4* driver. Ho et al. used the A53T mutant human α-synuclein protein under the *elav-Gal4* driver to develop flies with PD-like characteristics. These flies showed adult-onset loss of the dopaminergic neurons and locomotor dysfunction. Treatment with 3-hydroxybenzoic acid (3-HBA),

3,4-dihydroxybenzoic acid (3,4-diHBA), and 3-(3-hydroxyphenyl)propionic acid (3-HPPA), which are phenolic acid metabolites, inhibited α-synuclein aggregation in vitro, and improved the locomotor activity of PD model flies. It has been reported that *Bacteroides ovatus* could convert the flavanols catechin and epicatechin to DHCA, 3,4-diHBA, and 3-HBA. Moreover, *B. ovatus*, *Eggerthella lenta*, and *Escherichia coli* could produce 3-HPPA, 3,4-diHBA, and 3-HBA through metabolic processes independent of catechin and epicatechin. These findings imply that the gut microbiota might affect PD pathogenesis by modulating dietary flavanols [47].

### 6.3. PINK1 Mutant Fly Model

PTEN-induced kinase 1 (PINK1), a nuclear-encoded mitochondrial serine/threonine-protein kinase, has been associated with PD. In animal models with *PINK1* gene mutations, PD-associated changes and symptoms are observed, such as fragmented mitochondrial cristae, enhanced oxidative stress sensitivity, locomotor defects, and dopaminergic neuronal loss [113,120]. Therefore, PINK1 with loss-of-function mutations is used to develop PD model flies. Xu et al. showed that PD flies with PINK1 mutations exhibit reduced lifespan, climbing and flight defects, degenerated flight muscles, and loss of dopaminergic neurons. It has been shown that the compound epigallocatechin-3-gallate (EGCG), a major polyphenol present in green tea, can reduce PD symptoms. Moreover, EGCG could restore the microbiota diversity in PD flies by decreasing the gut abundance of *Proteobacteria* and increasing the gut abundance of *Firmicutes* and *Bacteroidetes*. Moreover, in PD flies, the gut abundance of the bacterium *Lactobacillus* and *Acetobacter* is increased, but in EGCG treatment groups, it reduced the abundance of *Acetobacter* and *L. plantarum*, and rescued the locomotor defects. Furthermore, increasing the abundance of *L. plantarum* in EGCG treated PD flies reduced the efficacy of this polyphenol. Thus, EGCG provides neuroprotection by affecting the gut microbiota in PD flies [48].

## 7. Autism Spectrum Disorder

Autism spectrum disorder (ASD) is a neurodevelopmental disorder that manifests in early childhood. It is characterized by complex behavioral phenotypes and deficits in both social and cognitive functions; however, the exact cause of ASD is still unknown [121,122]. In recent years, the association between genetic factors and ASD has been identified, including copy number variation, single nucleotide polymorphisms, and epigenetic alterations. Besides genetics, environmental factors, such as air pollutants and pesticides, also play a key role in ASD [122,123]. Recently, the *Drosophila* model has been used to study the relationship between ASD and chromatin remodeling, post-transcriptional regulation, protein synthesis/degradation, and cell adhesion molecules [124]. *POGZ* encodes the heterochromatin protein 1 α-binding protein that is critical for chromatin remodeling in humans. *POGZ* is believed to function as a transcriptional regulator, which is crucial for neuronal functions [125]. Downregulation of *row*, the *Drosophila* ortholog of *POGZ*, in neurons leads to a deficit in habituation, which is a form of non-associative learning [125]. Euchromatin histone methyltransferase (*EHMT*) is another example of the ASD risk gene related to chromatin remodeling [126,127,128]. Loss of *EHMT* in *Drosophila* leads to learning and memory deficits, and a significant decrease in dendritic end numbers, higher-order branching, and dendritic field complexity [129].

Recent studies suggest that the KDM5 family proteins are crucial to ASD. KDM5 family proteins are histone demethylases that regulate histone H3K4me3 modification, which is associated with promoters of transcriptionally active genes [130,131]. Loss-of-function mutations in KDM5 family proteins have been reported in patients with ASD [132,133]. Moreover, *kdm5c* knockout mice also exhibit abnormal learning and social behavior due to reduced spine density [134,135]. Flies carrying the mutant human *kdm5c* (*kdm5^A512P^*) exhibit learning and memory defects [136]. Furthermore, these flies exhibit behavioral defects, and transcriptional level changes, which are similar to a mutant allele that abolish demethylase activity. These studies suggest that KDM5 family proteins are critical for learning, memory, and social behaviors by their enzymatic function for gene activation [136].

There are several behavioral assays that can be used to study ASD in the *Drosophila* model. Social space assays were originally developed by Simon et al. to study ASD in adult flies [137]. This assay is carried out in a test chamber with two square glass plates (18 × 18 cm) separated by a 0.5 cm spacer, allowing the flies to be in a space. The internal space is triangular with a height and base of 15.3 cm each. Flies are placed at the bottom of the chamber, and all flies start climbing up from the same starting point at the bottom. After 20 min, the distance between the closest neighbors is measured using a camera [137]. Besides social space assays, the *Drosophila* activity assay is also commonly employed in the ASD model flies [138].

Chen et al. proposed that KDM5 proteins affect social behavior by regulating the gut microbiota composition [49]. In humans, the KDM5 family comprises of four members, whereas in *Drosophila*, only one KDM5 ortholog is present [139]. Chen et al. generated the *kdm5*-deficient flies and showed that loss of *kdm5* leads to intestinal barrier dysfunction and changes in social behavior. In addition, KDM5 deficiency in the intestinal tissues altered the microbiota composition, leading to gut barrier defects. However, the probiotic *Lactobacillus* or antibiotic administration to the *kdm5*-deficient flies partially rescued the behavioral and cellular phenotypes, and extended the lifespan (Table 1). Furthermore, KDM5 transcriptionally regulates the immune deficiency (IMD) signaling pathway, and maintains bacterial homeostasis in a demethylase-dependent manner [134]. These studies suggest that targeting the gut microbiota might be a possible clinical therapeutic approach for treating ASD patients with abnormal IMD signaling [49].

## 8. Conclusions

Since *Drosophila* harbors a simple gut microbial community, it is easy to understand the mechanisms by which the microbiota−gut−brain axis influences the complicated animal behaviors. Moreover, several fly models of neurodegenerative diseases have been developed allowing scientists to investigate the association between gut microbes and neuronal disorders, which could pave the way towards translational studies of probiotics.

This review emphasizes the importance of recent findings on the association between the gut microbiota and neuronal functions, behavior, and neurodegenerative diseases in the *Drosophila* model. The gut microbiota influences temperature tolerance, sleep behavior, locomotor activity, host-outbreed strategy, and aggressive behavior, all of which are innate behaviors. Although there are some contradictions regarding the association between the gut microbiota and behavior, we speculate that these differences might be due to different host genotypes and diets used in different studies. However, studies on *Drosophila* sleep and locomotor behaviors provide us with clues that the gut microbiota acts as neuromodulator of the neuronal circuits [34]. The association between learned behavior and gut microbiota remains poorly understood. Silva et al. reported that gut microbiota affects appetitive long-term memory, courtship learning, and courtship long-term memory [34]. Although the role of the microbiota−gut−brain axis in aging remains unclear, we learned from fruit flies that changes in lifestyle or diet could improve lifespan [40]. Moreover, studies on the fly model suggest that drinking milk with kefir and taking symbiotic formulations or probiotic supplements reduce AD symptoms [41,42,44]. The beneficial effects of probiotic supplements and EGCG containing green tea on PD symptoms have been shown in flies [116,118]. In future studies, the first priority will be to develop novel experimental setups that facilitate aseptic conditions for tight control and precise manipulation of the gut microbiota in fruit flies [34]. We believe that the *Drosophila* model can help us to understand the principle underlying the microbiota−gut−brain axis and identify the potential next-generation probiotics (NGP) in the near future.

## Figures and Tables

**Figure 1 biomedicines-10-00596-f001:**
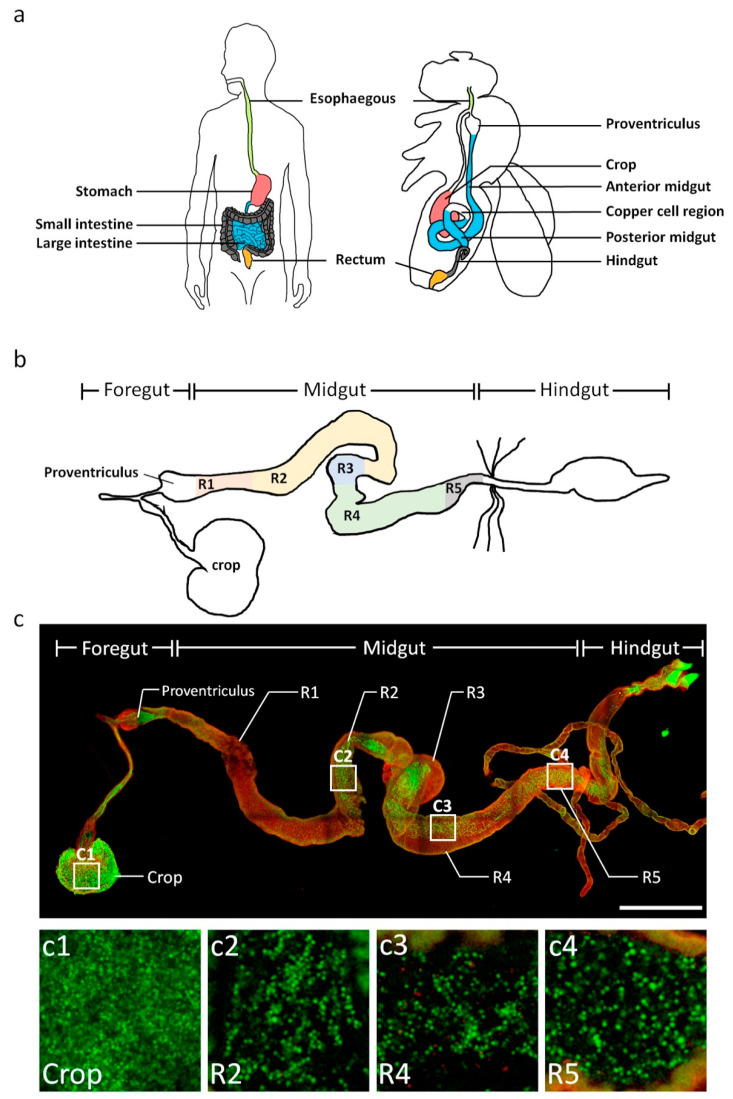
Comparison of human and *Drosophila* digestive tracts. (**a**) Digestive tracts of humans and *Drosophila*. The functional regions of the digestive tract in *Drosophila* and humans are similar and include esophagus, anterior and posterior midgut (small intestine), large intestine (hindgut), and stomach (crop). (**b**) Subregions of the *Drosophila* digestive tract depicting the foregut, midgut, and hindgut. Foregut contains crop and proventriculus. The midgut is further divided into R1-R5 regions. (**c**) GFP-tagged *Lactobacillus* was fed to adult *Drosophila* and confocal images show the distribution of *Lactobacillus* (green) in different regions of the digestive tract. The magnified images of crop, R2, R4, and R5 regions are showed in c1 to c4. The sample was immunostained with anti-integrin-βPS antibody (red). Scale bar, 500 μm.

**Table 1 biomedicines-10-00596-t001:** Microbiota related behaviors and neurodegenerative diseases in *Drosophila*.

Category		Related Microbiota	References
**Innate Behavior**			
Mating behavior			
	Sperm transfer	N.D.	[31]
	Fecundity	N.D.	[32]
Aggressive behavior		*Acetobacter*, *Lactobacillus*, *Enterococcus*	[33]
Sleep behavior		N.D.	[34]
Locomotion behavior		*Lactobacillus brevis*	[34,35,36]
Temperature tolerance		N.D.	[37,38]
**Learned Behavior**			
Olfactory memory			
	Long-term memory	N.D.	[34]
Courtship memory			
	Learning	N.D.	[34]
	Long-term memory	N.D.	[34]
**Aging**			
Longevity		*Lactobacillus plantarum*	[39,40]
**Alzheimer’s disease**			
*elav*-*Gal4*; *UAS*-*BACE*/*UAS*-*APP* model		Kefir related bacteria *Lactobacillus plantarum* NCIMB 8826 (Lp8826), *Lactobacillus fermentum* NCIMB 5221 (Lf5221), *Bifidobacteria longum spp. infantis* NCIMB 702255 (Bi702255)	[41,42]
*GMR*-*Aβ42* model		*Lactobacillus sakei* Probio65, *Lactobacillus paracasei* 0291 *Lactobacillus plantarum* DR7, *Stenotrophomonas*, *Acetobacter*, *Wolbachia*	[43,44]
*elav*-*Gal4*; *UAS*-*Aβ42* model		Acetobacteraceae, Lactobacillacea, *enterobacteria*	[45,46]
**Parkinson disease**			
*elav*-*Gal4*; *UAS*-*Synuclein* model		*Bacteroides ovatu*, *Eggerthella lenta*, *Escherichia coli*	[47]
*PINK1* mutant model		*Proteobacteria*, *Firmicutes*, *Bacteroidetes*, *Acetobacter*, *Lactobacillus*	[48]
**Autism spectrum disorder**			
*kdm5*-*deficient* model		*Lactobacillus plantarum*	[49]

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
