# Peer review of "Drosophila Model for Studying Gut Microbiota in Behaviors and Neurodegenerative Diseases"

_biomedicines, 2022, doi:10.3390/biomedicines10030596_

Round 1

Reviewer 1 Report

In this manuscript, the authors summarized the current understanding of the gut microbiota of Drosophila and its impact on behavior, physiology, and diseases; discussed the innate behaviors and learned behaviors in Drosophila; outlined the aging and longevity, Alzheimer’s diseases, Parkinson’s disease and autism spectrum disorder associated with gut microbiota in Drosophila; also discussed the next-generation probiotics and their potential applications. This study is overall well-designed, well written, and the literatures are well interpreted, and the logic discussion can be recognized as well. Few minor issues need to address:

  • The massive literatures have been well interpreted in the main text, however, the rationales for this study least described in the abstract. The authors may rewrite this section to fulfill readers.
  • All citing source is supposed to be cited accurately. The authors may need to go over all the references cited in the text to make sure they are cited properly. For example, reference 27 (PMID: 30356215) investigated locomotor behavior instead of courtship behavior, and the study reveals that the microbiota modulates walking speed and temporal patterns of locomotion in Drosophila which might be contrary to the perspective the authors described (At least three studies have reported that the gut microbiota does not affect courtship behavior in Drosophila - line 104).

Author Response

Reviewer1:

In this manuscript, the authors summarized the current understanding of the gut microbiota of Drosophila and its impact on behavior, physiology, and diseases; discussed the innate behaviors and learned behaviors in Drosophila; outlined the aging and longevity, Alzheimer’s diseases, Parkinson’s disease and autism spectrum disorder associated with gut microbiota in Drosophila; also discussed the next-generation probiotics and their potential applications. This study is overall well-designed, well written, and the literatures are well interpreted, and the logic discussion can be recognized as well. Few minor issues need to address:

  • The massive literatures have been well interpreted in the main text, however, the rationales for this study least described in the abstract. The authors may rewrite this section to fulfill readers.

Author’s response: Thank you very much for your valuable suggestions. We have added an appropriate description of the rationale for the study to the abstract as suggested. (Please refer to lines 28-32).

  • All citing source is supposed to be cited accurately. The authors may need to go over all the references cited in the text to make sure they are cited properly. For example, reference 27 (PMID: 30356215) investigated locomotor behavior instead of courtship behavior, and the study reveals that the microbiota modulates walking speed and temporal patterns of locomotion in Drosophila which might be contrary to the perspective the authors described (At least three studies have reported that the gut microbiota does not affect courtship behavior in Drosophila - line 104).

Author’s response: We have carefully checked all the references and insured they are cited properly. Thank you again for your valuable comments, which greatly strengthen the quality of our manuscript.

Reviewer 2 Report

The review article focused on developing Drosophila Model for analyze Gut Microbiota in Behaviors and Neurodegenerative Diseases. The topic is of clinical significance and very interesting. There were some suggestions:

  1. In the Introduction part, line 66-70 and also Fig 1: Please cite the reference.
  2. For Fig 1a, 1b, 1c, please describe it detail in the content.
  3. Also for Table 1, please describe it detail in the content.
  4. In the “Drosophila innate behaviors” part, line 82-89. Please briefly describe what the correlation is in the innate behavior between human and Drosophila? And what kind of innate behavior in human can be analyzed by the Drosophila model? Is there any examples or reference?
  5. Suggest briefly describe “Gut Microbiota and Behaviors and Neurodegenerative Diseases in human” before “2. Drosophila innate behaviors”(line 81)
  6. As for “2.1. Gut Microbiota and Mating Behavior; 2.2. Gut Microbiota and Aggressive Behavior; 2.3. Gut Microbiota and Sleep and Locomotor Activity. 2.4. Gut Microbiota and Temperature Related Behaviors” line 92-229. I am not sure what is the key-point in this part? As the title of this article is “Drosophila Model for Studying……” The most important is to use drosophila as a model to analyze neurologic or psychiatric disease but not describe the behavior of Drosophila. Suggest revise this part to focus on the topic.
  7. The same for “3. Drosophila learned behaviors”, “4. Aging and Longevity” “5. Alzheimer’s Diseases”, “6. Parkinson’s Disease”, “7. Autisum Spectrum Disorder” line 230-527. The content is too busy, confusing and not easy to read. Suggest describe human disease first and then focus on the progress in the use of Drosophila model to analyze such disease. Some drosophila behavior without the correlation with human should be deleted.
  8. As for “Next Generation Probiotics and Their Potential Applications” line 527-548. I am not sure the correlation between this part with the title?

In conclusion since the title is”Drosophila Model for Studying……”. The key-point is “use Drosophila as a model” but not describe the behavior of drosophila.

Author Response

Reviewer 2:

The review article focused on developing Drosophila Model for analyze Gut Microbiota in Behaviors and Neurodegenerative Diseases. The topic is of clinical significance and very interesting. There were some suggestions:

1. In the Introduction part, line 66-70 and also Fig 1: Please cite the reference.

Author’s response: Per your suggestion, we have cited the relevant references. (Please see line 74).

2. For Fig 1a, 1b, 1c, please describe it detail in the content.

Author’s response: The detailed descriptions of Fig. 1 are shown in lines 77-85.

3. Also for Table 1, please describe it detail in the content.

Author’s response: Table 1 is the summary of the relationship between gut microbiota and Drosophila behaviors as well as human related neurodegenerative diseases, which have been mentioned in the manuscript. Therefore, the detailed descriptions of each behavior and disease, and their related microbiota are shown in the text of the manuscript.

4. In the “Drosophila innate behaviors” part, line 82-89. Please briefly describe what the correlation is in the innate behavior between human and Drosophila? And what kind of innate behavior in human can be analyzed by the Drosophila model? Is there any examples or reference?

Author’s responses: Thank you very much for your insightful suggestions. We have clearly discussed the correlation of innate behavior between human and Drosophila, and also added the relevant references as suggested. Please refer to lines 91-98.

5. Suggest briefly describe “Gut Microbiota and Behaviors and Neurodegenerative Diseases in human” before “2. Drosophila innate behaviors”(line 81)

Author’s responses: We have concisely described the “Gut Microbiota and Behaviors and Neurodegenerative Diseases in human” in the introduction section (lines 58-68).

6. As for “2.1. Gut Microbiota and Mating Behavior; 2.2. Gut Microbiota and Aggressive Behavior; 2.3. Gut Microbiota and Sleep and Locomotor Activity. 2.4. Gut Microbiota and Temperature Related Behaviors” line 92-229. I am not sure what is the key-point in this part? As the title of this article is “Drosophila Model for Studying……” The most important is to use drosophila as a model to analyze neurologic or psychiatric disease but not describe the behavior of Drosophila. Suggest revise this part to focus on the topic.

Author’s response: The title of this review is “Drosophila Model for Studying Gut Microbiota in Behaviors and Neurodegenerative Diseases”. Given the importance of microbiota in the orchestration of behavior and neurological diseases, we therefore not only illustrated the impact of microbiota on neurodegenerative diseases but also comprehensively discussed animal behaviors in the article.     

7. The same for “3. Drosophila learned behaviors”, “4. Aging and Longevity” “5. Alzheimer’s Diseases”, “6. Parkinson’s Disease”, “7. Autisum Spectrum Disorder” line 230-527. The content is too busy, confusing and not easy to read. Suggest describe human disease first and then focus on the progress in the use of Drosophila model to analyze such disease. Some drosophila behavior without the correlation with human should be deleted.

Author’s response: Thank you for your kind suggestions. We have clearly described the human diseases before the Drosophila model to analyze diseases in the manuscript. Please see lines 314-319, 366-379, 439-448, and 494-502.

8. As for “Next Generation Probiotics and Their Potential Applications” line 527-548. I am not sure the correlation between this part with the title?

Author’s response: Thank you for your comments. We have deleted all this section to keep all the content relevant to the title of the manuscript.

In conclusion since the title is ”Drosophila Model for Studying……”. The key-point is “use Drosophila as a model” but not describe the behavior of drosophila.

Author’s response: Thank you for your suggestions. We believe the background introduction of Drosophila behaviors is important since most of the readers are not familiar with the fly behaviors. We therefore keep the descriptions of fly behaviors and their related assay systems in the manuscript. We sincerely thank the reviewer for the valuable comments, which greatly improve the quality of the review article.

Round 2

Reviewer 2 Report

The authors had revised the article according to the suggestions